

# HPV molecular detection from urine versus cervical samples: an alternative for HPV screening in indigenous populations

Francisco I. Torres-Rojas[1,*], Miguel A. Mendoza-Catalán[1,*], Luz del C. Alarcón-Romero[2], Isela Parra-Rojas[3], Sergio Paredes-Solís[4], Marco A. Leyva-Vázquez[1], Jair E. Cortes-Arciniega[1], Carlos J. Bracamontes-Benítez[1] and Berenice Illades-Aguiar[1]

[1] Laboratorio de Biomedicina Molecular. Facultad de Ciencias Químico Biológicas, Universidad Autónoma de Guerrero, Chilpancingo de los Bravo, Guerrero, Mexico
[2] Laboratorio de Citopatología e Histoquímica. Facultad de Ciencias Químico Biológicas, Universidad Autónoma de Guerrero, Chilpancingo de los Bravo, Guerrero, Mexico
[3] Laboratorio de Investigación en Obesidad y Diabetes, Facultad de Ciencias Químico Biológicas, Universidad Autónoma de Guerrero, Chilpancingo, Guerrero, México
[4] Centro de Investigación de Enfermedades Tropicales, Universidad Autónoma de Guerrero, Acapulco, Guerrero, México
* These authors contributed equally to this work.

Corresponding author
Berenice Illades-Aguiar,
billades@uagro.mx

## ABSTRACT

**Background.** Cervical cancer (CC) is the fourth leading cause of death from neoplasms in women and is caused by the human papilloma virus (HPV). Several methods have been developed for the screening of cervical lesions and HPV; however, some socio-cultural factors prevent women from undergoing gynecological inspection, which results in a higher risk of mortality from cervical cancer in certain population groups as indigenous communities. This study aimed to compare the concordance in HPV detection from urine and cervical samples, to propose an alternative to cervical scraping, which is commonly used in the cervical cancer screening.

**Methodology.** The DNA from cervical scrapings and urine samples was extracted using the proteinase K method followed by precipitation with alcohol, phenol andchloroform; a modification of the proteinase K method was developed in the management of urine sediment. Viral genotyping was performed using INNOLipa.

**Results.** The study population consisted of 108 patients from an indigenous population at southern Mexico, 32 without squamous intraepithelial lesions (NSIL) and 76 with low squamous intraepithelial lesions (LSIL). The majority of NSIL cervical scrapes were negative for HPV (90.63%), whereas more than half of LSIL cases were high-risk HPV positive (51.32%), followed by multiple infection by HR-HPV (17.11%), and multiple infection by LR- and HR-HPV (9.21%). No statistically significant relationship between the cytological diagnosis and the HPV genotypes detected in the urine samples was observed. A concordance of 68.27% for HPV positivity from urine and cervical samples was observed. Similarly, a concordance of 64.52% was observed in the grouping of HPVs by oncogenic risk. HR-HPV was detected in 71% of the urine samples from women with LSIL diagnosis, which suggests that HR-HPV detected in a urine sample could indicate the presence or risk of developing SIL.

**Conclusion**. HR-HPV detection in urine samples could be an initial approach for women at risk of developing LSIL and who, for cultural reasons, refuse to undergo a gynecological inspection.

## INTRODUCTION

Cervical cancer (CC) is the third most frequent neoplasm in women in the world and the second most frequent among those who are 15–44 years old (*Arbyn et al., 2020*). Almost all CC cases are related to high-risk human papillomavirus (HR-HPV) infection, with HPV16 and HPV18 being the most common HR-HPVs in CC (*Téguété et al., 2017*; *Chan et al., 2019*). The progression of cervical cancer involves premalignant transition stages, which are known as squamous intraepithelial lesions of low or high degree according to the Bethesda System (LSIL or HSIL, respectively) (*Nayar & Wilbur, 2015*). Multiple HPV infection is very common in the diagnosis of precursor lesions. It has also been proposed that diverse HPVs can develop synergism in the cell, which could be associated with the progress of lesions (*Sohrabi & Hajia, 2017*).

Clinical diagnosis is focused on the cellular morphological changes caused by HPV infection. The Papanicolaou test is considered the gold standard for early detection of cervical carcinoma, however, the Pap test results should not be considered to be a determinant criterion in the treatment decision (*Sayyah-Melli et al., 2019*; *Kitchen & Cox, 2020*). It is highly recommended that the cytological diagnosis via the Pap test be complemented with molecular HPV detection to increase the certainty of the diagnosis and maximize cancer prevention.

Samples for molecular HPV detection must be collected by trained personnel (*Mittal & Yadav, 2019*); however, in some populations, such as the Nahuatl in Mexico, collection of this type of sample by medical personnel is often not allowed (*Graham & Mishra, 2011*; *Giorgi-Rossi, Baldacchini & Ronco, 2014*). Hence, it is necessary to propose alternatives for sampling. In Mexico, the cervical cancer screening program is focused on the detection of cervical premalignant lesions using the Pap test, in sexually active women from 25 to 34 years old, and HPV detection in women from 35 to 64 years old with repetition every 5 years when the Pap test is negative (*Centro Nacional de Equidad de Género y Salud Reproductiva, 2015*). Although this service is free of charge in any institution of the National Health System, many women, mainly those who belong to indigenous communities, refuse a gynecological inspection for cultural reasons. Therefore, it is highly important to identify less invasive screening alternatives to include this type of population. The proposed options to replace the Pap test include the self-sampling method (*Dzuba et al., 2002*); sampling through cervicovaginal washes (*Nobbenhuis et al., 2002*); and, interestingly, urine, which, in recent reports has been detailed as a useful alternative for HPV detection (*Pattyn et*

*al., 2019b*; *Lefeuvre et al., 2020*). Most studies focused on HPV detection from urine have worked with the first urine of the day and the extraction of DNA using commercial kits (*Khunamornpong et al., 2016*; *Nilyanimit et al., 2017*). To date, no studies have investigated the Nahuatl indigenous community, whose sociodemographic characteristics such as gender perspectives and low access to health systems (*Leyva-Flores et al., 2013*), make it vulnerable to HPV infection. In adittion, data relating to the level of concordance in the diagnostic tests between cervical scraping and urine are scarce. To address the above, this work aims to determine the concordance between the molecular detection of HPV in Nahuatl from urine and cervical scraping samples, the latter being considered the gold standard, using an efficient and low-cost method for DNA extraction from urine samples.

## MATERIALS AND METHODS

### Study population
This study was conducted in the Nahuatl indigenous communities of Xalitla, San Juan Tetelcingo, San Agustín Oapan, San Miguel Tecuixiapan, and Ahuehuepan in the municipality of Tepecuoacuilco de Trujano, which is located in the northern part of the state of Guerrero, Mexico. Indigenous women were invited to participate in the study by calling with loudspeakers in the Nahuatl language and subsequently visiting homes. The invitation was for women to attend their community health center, to have a Pap smear, for the molecular detection of HPV in cervical scrapings and urine samples. The women who attended the health center were surveyed to determine their age, schooling, gynecological-obstetrical background, knowledge of cervical cancer, and whether they had had a Pap smear. The visit to the communities and the recruitment of the women who decided to participate in the study took place from September to November 2019. The research protocol was approved by the Ethics Committee of the Universidad Autonóma de Guerrero under identification number 03/07/2019 and the participating women signed informed consent.

### Specimen collection and preparation
The women included in the study had an exo-endocervical sample taken for the Pap smear and cervical HPV detection. The cervical specimen was collected using a cervix brush directed at the transformation zone (TZ) and an Ayre wooden spatula for ectocervix zone sampling of the uterine cervix and placed into PreservCyt solution (Cytyc Corporation, Marlborough, MA) for liquid-based cytology. In addition, another cervical specimen for HPV detection was collected using a Dacron swab and placed in universal collection medium (UCM) (Digene Corporation, Gaithersburg, MD). Both samples were transferred to Facultad de Ciencias Químico Biológicas (FCQB) and stored at room temperature, until the Pap smear and HPV-molecular detection. In addition, the patients were asked for a sample from the first-void urine, collected themselves in a 10 x 60 mL plastic urine collection specimen bottle, stored at 4 °C for a maximum period of 72 h, and transferred to FCQB. A quantity of 15 ml of urine was centrifuged at 3500 rpm for 20 min and then washed with PBS 1X twice, and DNA extraction and HPV detection/genotyping were subsequently performed. Another fraction of the sample was used for dipstick urinalysis.

## Dipstick urinalysis

Dipstick urinalysis was performed using Combur 10-Test M strips according to manufacturer instructions. Briefly, 10–15 mL was taken from the specimen container and one strip was submerged for 15–30 s; then, the strip was read with reference to the scale printed on the packaging. The strips had reagent pads for the semiquantitative assessment of density, pH, leukocyte esterase, nitrite, protein, glucose, ketones, urobilogen, bilirubin, and hemoglobin/myoglobin.

## Cytological diagnosis

The cytological diagnosis of the exo-endocervical samples was performed according to the Bethesda System (*Solomon et al., 2002*; *Nayar & Wilbur, 2015*). Slides with the cytological smears of the TZ for conventional cytology examination were fixed in ethanol for 10 min. The slides were then stained using the Papanicolaou kit (cat. no. 64294; Hycel, Chemical Reagents). Briefly, the slides were hydrated in a descending alcohol series and then incubated at room temperature for 45 s with Harris hematoxylin to stain the nuclei. Additionally, Orange G colorant was added and incubated at room temperature for 80 s, followed by EA-50 incubated at room temperature for 3 min, which stained the eosinophils and basophils cells, respectively. The slides were then cleared with Xylol reagent prior to microscopic observation (DM1000 LED; Leica Microsystems, Inc.; magnification, 10x–20x).

Alternatively, the samples for liquid-based cytology were processed according to the manufacturer's protocol of liquid-PREP$^{TM}$ (LGM International, Inc.). Briefly, a clearing solution was added to each sample and then the samples were centrifuged at $1000 \times$ g for 5 min at room temperature. The supernatant was discarded after the addition of the cell base solution, which conserved the pellet. The samples were mixed and 10 $\mu$l was added to the slide, which was fixed at room temperature with ethanol for 10 min, following by staining using the Papanicolaou kit and microscopic observation (DM1000 LED; Leica Microsystems, Inc.; magnification, 10x–20x).

## DNA extraction and integrity

DNA was extracted from cervical scrapes via the phenol-chloroform-isoamyl alcohol method (*Ausubel et al., 1995*) following proteinase K digestion at 64 °C for 45 min. For the extraction of DNA from the urine pellet, proteinase K was added, and then the sample was incubated in a water bath for 15 min and subsequently incubated at 55 °C overnight. Finally, the phenol-chloroform-isoamyl alcohol procedure was performed. The DNA recovered was diluted in DEPC water and and quantified by spectrophotometry. Although a DNA purity (260/230 nm absorbance ratio) $\geq$ 1.6 is suggested for an optimal PCR result (*Gallagher, 2001*), we considered a DNA purity $\geq$1.4 as suitable for viral genotyping, considering the DNA integrity determined through β-globin gene detection, thereby, 108 samples were considered to be viable (Fig. S1).

## HPV detection and genotyping

The DNA was subjected to an HPV genotyping assay using an INNO-LiPA HPV Genotyping Extra II assay (INNO-LiPA; Fujirebio Europe, Ghent, Belgium) according

to the manufacturer's instructions. This system amplifies a 65-bp fragment of the L1 open reading frame and enables the identification of 32 HPV genotypes including 13 high-risk types (HPV 16, 18, 31, 33, 35, 39, 45, 51, 52, 56, 58, 59, and 68), six probable high-risk types (HPV 26, 53, 66, 70, 73 and 82), and thirteen low-risk or unknown risk types (HPV 6, 11,40, 42, 43, 44, 54, 61, 62, 67, 81, 83 and 84). For data analysis, the HPVs detected in the samples were grouped according to oncogenic risk, as HR-HPV, LR-HPV, multiple infection by LR- and HR-HPV (MI LR/HR-HPV), and unidentified HPV infection (HPV-X).

## Statistical analysis

Data capture and statistical analysis were performed with the statistical program STATA 14.0 (College Station, TX: StataCorp LP). Qualitative variables were analyzed by chi-square test ($X^2$) or Fisher's exact test as appropriate. In quantitative variables, the Shapiro–Wilk test was carried out for normality determination and the Mann–Whitney test was applied for comparison between two groups. Finally, the sensitivity, specificity, and concordance (Cohen's Kappa coefficient) were calculated, considering HPV detection from cervical scrapings as the reference. A Kappa value of less than 0.20 indicates poor agreement, 0.21–0.40 moderate agreement, 0.61–0.80 good agreement and 0.81–1.00 very good agreement. $p$ values ≤ 0.05 were considered statistically significant (*Brennan & Silman, 1992*).

# RESULTS

After carrying out sensitization in the indigenous communities, 155 indigenous women attended the health centers and were surveyed: 24 from Ahuehuepan (20.34% of the total number of attendees), 30 from San Agustín Oapan (27.03%), 29 from San Juan Tetelcingo (40.28%), 41 from San Miguel Tecuixiapan (44.09%) and 31 (25.85%) from Xalitla of the municipality of Tepecoacuilco, Guerrero, Mexico. The median age was 47.5 years, 62% of the women had not gone to school, 62% did not consume alcohol, 81% had had a Pap smear but mentioned that on many occasions they had not been given the results, and more than 90% did not know about cervical cancer (data not shown). Of the 155 women surveyed, 108 appropriate cervical scrapings and 104 urine samples were obtained. This difference occurred because some of the women did not agree to undergo a cervical scraping (7.74%), others were menstruating (4.52%), and some of the urine samples did not have the appropriate DNA purity for processing (18.06%).

Cytological analysis showed that 32 women did not have squamous intraepithelial lesions (NSIL) and 76 had low squamous intraepithelial lesions (LSIL). The median ages of the women were 49.0 (39.0–55.0) and 45.5 (29.0–56.5) years in the NSIL and LSIL groups, respectively. The clinical and pathological characteristics are summarized in Table 1. In this population, only a greater number of sexual partners showed a statistically significant relationship with LSIL (Table 1).

## HPV positivity in cervical and urine samples

As expected, based on analysis the cervical samples, a statistically significant relationship was observed between the HPV-oncogenic risk detected and cytological diagnosis ($p = 0.001$). The NSIL samples were significantly negative to HPV (90.63%), whereas LSIL

**Table 1  Clinical characteristics of the indigenous population studied.**

| | Cytological diagnosis | | | p value* |
|---|---|---|---|---|
| | NSIL<br>n = 32<br>n (%) | LSIL<br>n = 76<br>n (%) | Total<br>n = 108<br>n (%) | |
| **Age (years)**[a] | 49 (39–55) | 45.5 (29–56.5) | 47.5 (32–55.5) | 0.28[b] |
| **Alcohol consumption** | | | | |
| No | 19 (29.69) | 45 (70.31) | 64 (59.26) | 0.98[c] |
| Yes | 13 (29.55) | 31 (70.45) | 44 (40.74) | |
| **Sexual partners** | | | | |
| One | 27 (38.03) | 44 (61.97) | 71 (66.36) | 0.013[d] |
| ≥Two | 5 (13.89) | 31 (86.11) | 36 (33.64) | |
| **Menarche** | | | | |
| ≤14 years old | 27 (28.72) | 67 (71.28) | 94 (87.04) | 0.23[d] |
| ≥15 years old | 5 (35.71) | 9 (64.29) | 14 (12.96) | |
| **Number of births** | | | | |
| None | 4 (21.05) | 15 (78.95) | 19 (17.59) | |
| One | 2 (16.67) | 10 (83.33) | 12 (11.11) | 0.41[d] |
| ≥Two | 26 (33.77) | 51 (66.23) | 77 (71.30) | |
| **Abortions** | | | | |
| None | 25 (31.25) | 55 (68.75) | 80 (74.07) | |
| One | 7 (38.89) | 11 (61.11) | 18 (16.67) | 0.08[d] |
| ≥Two | 0 (0) | 10 (100.00) | 10 (9.26) | |
| **C-section** | | | | |
| None | 21 (25.30) | 62 (74.70) | 83 (76.85) | |
| One | 4 (33.33) | 8 (66.67) | 12 (11.11) | 0.10[d] |
| ≥Two | 7 (53.85) | 6 (46.15) | 13 (74.08) | |
| **Pap previous** | | | | |
| Yes | 27 (29.35) | 65 (70.65) | 92 (85.19) | 1.00[d] |
| No | 5 (31.25) | 11 (68.75) | 16 (14.81) | |

**Notes.**

NSIL, non-squamous intraepithelial lesions; LSIL, low-grade squamous intraepithelial lesions; C-section, caesarean section.

[a]Median (p25–p75).

[b]Mann Whitney-test.

[c]$X^2$ test.

[d]Fisher's exact test.

cases were positive to HR-HPV (51.32%), followed by multiple infection by HR-HPV (17.11%), and multiple infection by LR- and HR-HPV (9.21%), (Table 2). By contrast, no statistically significant relationship between the cytological diagnosis and the HPV genotypes detected in the urine samples was observed ($p = 0.33$). The most frequent HR-HPV genotypes detected in cervical samples were HPV-52 (10/108 cases), followed by HPV-58 and HPV-59 (9/108 cases), whereas the most frequent HR-HPV genotypes in urine samples were HPV-39 (21/105), followed by HPV-16 (17/105), HPV-52 (16/105), and HPV-51 (15/105) (Table S1). The number of HPV genotypes detected in cervical and urine samples was associated to cytological diagnosis ($p = 0.001$ and $p = 0.047$, respectively); In

**Table 2 HPV infection detected in cervix and urine samples.**

| | Cytological diagnosis | | | p value[*] |
| --- | --- | --- | --- | --- |
| | NSIL n (%) | LSIL n (%) | Total n (%) | |
| **HPV infection in cervix** | | | | |
| Negative | 29 (90.63) | 3 (3.95) | 32 (29.63) | |
| LR-HPV | 0 (0) | 3 (3.95) | 3 (2.78) | |
| HR-HPV | 1 (3.13) | 39 (51.32) | 40 (37.04) | |
| MI LR/HR-HPV | 2 (3.13) | 20 (9.21) | 22 (7.41) | 0.001[c] |
| HPV-X | 0 (0) | 11 (14.47) | 11 (10.19) | |
| **HPV infection in urine** | | | | |
| Negative | 9 (28.13) | 10 (13.70) | 19 (18.10) | |
| LR-HPV | 1 (3.13) | 3 (4.11) | 4 (3.81) | |
| HR-HPV | 10 (31.25) | 18 (24.66) | 28 (26.67) | 0.306[c] |
| MI LR/HR-HPV | 10 (15.63) | 34 (13.70) | 44 (14.29) | |
| X-HPV | 2 (6.25) | 8 (10.96) | 10 (9.52) | |
| **HPV genotypes in cervix** | | | | |
| None | 29 (90.63) | 3 (3.95) | 32 (29.63) | |
| One | 1 (3.13) | 53 (69.74) | 54 (50.00) | 0.0001[c] |
| ≥two (MI) | 2 (6.25) | 20 (26.32) | 22 (20.37) | |
| **HPV genotypes in urine** | | | | |
| None | 10 (31.25) | 9 (12.50) | 19 (18.27) | |
| One | 13 (40.63) | 29 (40.28) | 42 (40.38) | 0.047[c] |
| ≥two (MI) | 9 (28.13) | 34 (47.22) | 43 (41.35) | |

**Notes.**

NSIL, non-squamous intraepithelial lesions; LSIL, low-grade squamous intraepithelial lesions; HPV-LR, low risk HPV infection; HPV-HR, high risk HPV infection; MI LR/HR-HPV, low risk HPV and high risk HPV multiple infection; HPV-X, unidentified HPV; MI, multiple infection.

[c]Fisher's exact test.

[*]p value ≤ 0.05 was considered as statistically significant

addition, HR-HPV was detected in 71% (HR-HPV + MI LR/HR-HPV + MI HR-HPV) of urine samples from women with LSIL diagnosis (Table 2).

## Concordance between urine and cervical samples for HPV detection

A positive agreement of 86.11% for HPV determined by INNOLiPA from urine and cervical samples was observed, regardless of the viral genotype, however, the total concordance was poor or weak (Kappa = 0.16, concordance 68.27%) (Table 3). A similar result was observed by grouping the detected HPVs by oncogenic risk; an agreement of 89.6% in HR-HPV positivity between urine and cervical samples was observed (Kappa = 0.16, concordance 64.52%) (Table 4). The sensitivity and specificity of HR-HPV positivity in urine, using cervical scrapings as reference, were 89.7% and 25.7%, respectively.

## DISCUSSION

Currently, cervical cancer (CC) is the third most frequent neoplasm in women in the world (*Arbyn et al., 2020*), and nearly 100% of CC cases are related to high-risk human
**Table 3  Concordance, sensitivity, and specificity of HPV detection in urine and cervical samples.**

| HPV detection in cervical scraping | Urine HPV detection | | Total |
|---|---|---|---|
| | **HPV positive** | **HPV negative** | **n = 104** |
| HPV positive | 62 (86.11) | 10 (13.89) | 72 (69.23) |
| HPV negative | 23 (71.88) | 9 (28.12) | 32 (30.77) |
| | 85 (81.73) | 19 (18.27) | |
| | Sensitivity: 86.10% | | |
| | Specificity: 28.10% | | |
| | Concordance: 68.27% | | |
| | Kappa:0.16 | | |

**Table 4  Concordance, sensitivity, and specificity in detection of HPV-oncogenic risk groups in urine and cervical samples.**

| HPV detection in | Urine HPV detection | | | Total |
|---|---|---|---|---|
| Cervical scraping | **HR-HPV** | **LR-HPV** | **HPV negative** | **n = 93** |
| HR-HPV | 52 (89.66) | 0 (0) | 6 (10.37) | 58 (62.37) |
| LR-HPV | 2 (66.67) | 1 (33.33) | 0 (0) | 3 (3.23) |
| Negative | 24 (75.00) | 0 | 8 (25.00) | 32 (34.40) |
| | 78 (83.87) | 1 (1.08) | 14 (15.05) | |
| | Sensitivity: 89.70% | | | |
| | Specificity: 25.70% | | | |
| | Concordance: 64.52% | | | |
| | Kappa:0.16 | | | |

**Notes.**
HR-HPV, high risk HPV infection; LR-HPV, low risk HPV infection; HR-HPV, high risk HPV infection; LR-HPV, low risk HPV infection.

papillomavirus (HR-HPV) infection (*Hooi et al., 2018*; *Chan et al., 2019*). In developing countries, cancer mortality and morbidity are higher than those in developed countries, and an elevated cervical cancer-related mortality has been reported mainly in indigenous women. This suggests structural, social, or individual barriers to screening contribute to the poor prognosis of cancer cases in indigenous women (*Cramb et al., 2012*; *Vasilevska et al., 2012*).

The factors determining the high prevalence of cervical cancer in Mexico, particularly in Guerrero State, include the socio-cultural characteristics of the population, lack of marital support for screening, cultural taboos, stigmatization of women with this neoplasm, and, finally, the limited information regarding the procedure for early diagnosis of HPV infection (*Nilyanimit et al., 2017*). In addition, indigenous populations have low detection coverage, and many indigenous women refuse gynecological inspection due to cultural barriers, such as shame, or prohibition by their husband or other women in the community. These factors block timely HPV detection and increase the risk of developing premalignant lesions.

Although gynecological inspection via the Papanicolaou test and HPV molecular detection are the most important methods for screening of premalignant lesions of cervical

cancer, it is important to identify less invasive alternatives to obtain useful samples for HPV detection. Therefore, this study evaluated the concordance of HPV molecular detection from two different samples, cervical scraping, and urine, to evaluate urine as a potential alternative for screening in indigenous communities.

In the indigenous population studied, only the number of sexual partners was found to show a relationship with LSIL (Table 1), which is in agreement with previous reports that indicate a close relationship of this factor with the development of cervical neoplasm (*Herrero et al., 1990*; *Itarat et al., 2019*). Other factors such, as parity and age of first sexual encounter, have been related to premalignant lesions and cervical cancer development (*Lukac et al., 2018*; *Kashyap et al., 2019*).

Interestingly, in this study population HPV-52 was most frequent detected in cervical scrapings. This finding did not agree with previous reports in the region indicating that HPV-16 and HPV-18 are most frequent (*Illades-Aguiar et al., 2009*; *Illades-Aguiar et al., 2010*). Previous findings indicate differences in the HPV distribution between ethnic groups (*Lin et al., 2015*; *Baloch et al., 2017*). In this regard, is important to note that the analyzed population belongs to a socioeconomic zone with a high of migration rate (about 78%) and this situation can cause changes in the prevalence of sexually transmitted infections in the community (*World Health Organization, 2003*; *Platt et al., 2013*). One of the most important characteristics in the progression of cervical lesions is the presence of several viral genotypes. Thus, a finding of this study that should be noted, as shown in Table 2, is the high prevalence of multiple infections in the analyzed groups, which is a common finding in precursor lesions, as previously reported (*Schmitt et al., 2013*; *Aguilar-Lemarroy et al., 2015*).

The potential HPV detection from urine has been previously described by various authors; however, most have suggested performing DNA extraction using commercial kits (*Brinkman et al., 2002*; *Cuzick et al., 2017*; *Pattyn et al., 2019b*), which would increase the cost of the test and the difficulty of using urine samples as a feasible option for HPV screening. In this study, the DNA was extracted from both type of samples using the common phenol-chloroform-isoamyl alcohol method. Thus, DNA with adequate quality was obtained for molecular HPV detection, which was determined by reviewing the DNA integrity via β-globin gene detection. Molecular HPV detection from urine and cervical scraping samples may be similar in terms of cost, however, the importance of the use of urine samples is that it is a non-invasive method, and could be a useful alternative for women who refuse the gynecological inspection.

Different HPV genotypes were detected in urine samples compared to cervical scrapings; for instance, HPV 16 and HPV 39 were more frequent in urine in comparison with cervical samples (Table S1). Similarly, in a study in Chilean women, no significant differences in HPV detection and genotyping between the cervical and urine samples were observed. In some cases, the detection of carcinogenic HPV was positive in the cervical but negative in the urine samples, whereas in a similar number of cases, samples were positive for HR-HPV detection in urine but negative in cervical samples (*Buchegger et al., 2018*). In addition, Tanzi and colleagues (*2013*) reported that the absolute number of genotypes detected in urine samples was higher than the number of genotypes identified following examination

of cervical samples. Other studies have reported that HPV prevalence was similar or higher in another genital regions compared to cervical samples; it was observed that prevalence for any HPV type in vaginal specimens was greater than that in cervical samples, whereas the prevalence for any carcinogenic HPV type in vaginal and cervical specimens was similar, suggesting carcinogenic HPV genotypes could have similar tropism for vaginal and cervical epithelium (*Castle et al., 2007*). One possible explanation for this observation is that HPV of other anatomic sites, such as the urethra, vulva or vagina could be present in the urine sample considering the natural route of this liquid waste (*Sehgal et al., 2009*; *Tanzi et al., 2013*; *Abelson et al., 2018*; *Buchegger et al., 2018*).

A concordance of 68.27% in HPV detection from urine and the cervix was found (Table 3). In addition, one or more HR-HPVs was detected in 71% of urine samples from women with LSIL diagnosis (Table 2). The above findings suggest that two or more viral genotypes, mainly HR-HPV, detected in a urine sample could indicate the presence of, risk of developing, SIL. Therefore, urine can be a suitable sample in populations of women who do not accept gynecological inspection to obtain cervical scrapings. Other studies have supported the idea of using urine for HPV detection (*Brinkman et al., 2002*; *Sargent et al., 2019*; *Pattyn et al., 2019a*; *Tranberg et al., 2020*).

A limitation of this study is that there were only two study groups, NSIL and LSIL. With a larger population and at least three study groups (NSIL, LSIL, and HSIL), the concordance in the HPV molecular detection by INNOLiPA from urine and cervical samples could be more evident. Considering the experience gained in this study, in addition to that reported by other relevant studies, we propose a workflow with urine samples for the detection of HPV, focused mainly on HR-HPV, and the possible management of patients (Fig. 1). According to the guidelines of the Mexican program for the timely detection of cervical cancer, after a negative result of HPV in cervical scrapings, the test should be conducted every five years (*Centro Nacional de Equidad de Género y Salud Reproductiva, 2015*). In this workflow for HPV screening from urine samples, considering that sensibility of HPV detection from urine samples was 89.7% in relation to the molecular detection of HPV from cervical scrapings, we proposed that women aged 35–64 years old with a negative result of HR-HPV or positive result of LR-HPV could be considered to be at low risk and their next analysis could be scheduled after four and three years, respectively. In contrast, women who test positive for HR-HPV could be considered to be at risk of developing squamous intraepithelial lesions of the cervix (SIL) and should be submitted to a Pap test for gynecological inspection to confirm or rule out the presence of lesions and infection by HPV in cervical cells (Fig. 1).

Although the Pap test continues to be the gold standard for the determination of cervical abnormalities, the molecular detection of HR-HPV in urine samples is a non-invasive method. Thus, it could represent an initial approach for women at risk of developing SIL and a feasible alternative for indigenous women who, due to cultural barriers and poor health services, do not have timely detection of pre-malignant lesions and cervical cancer.

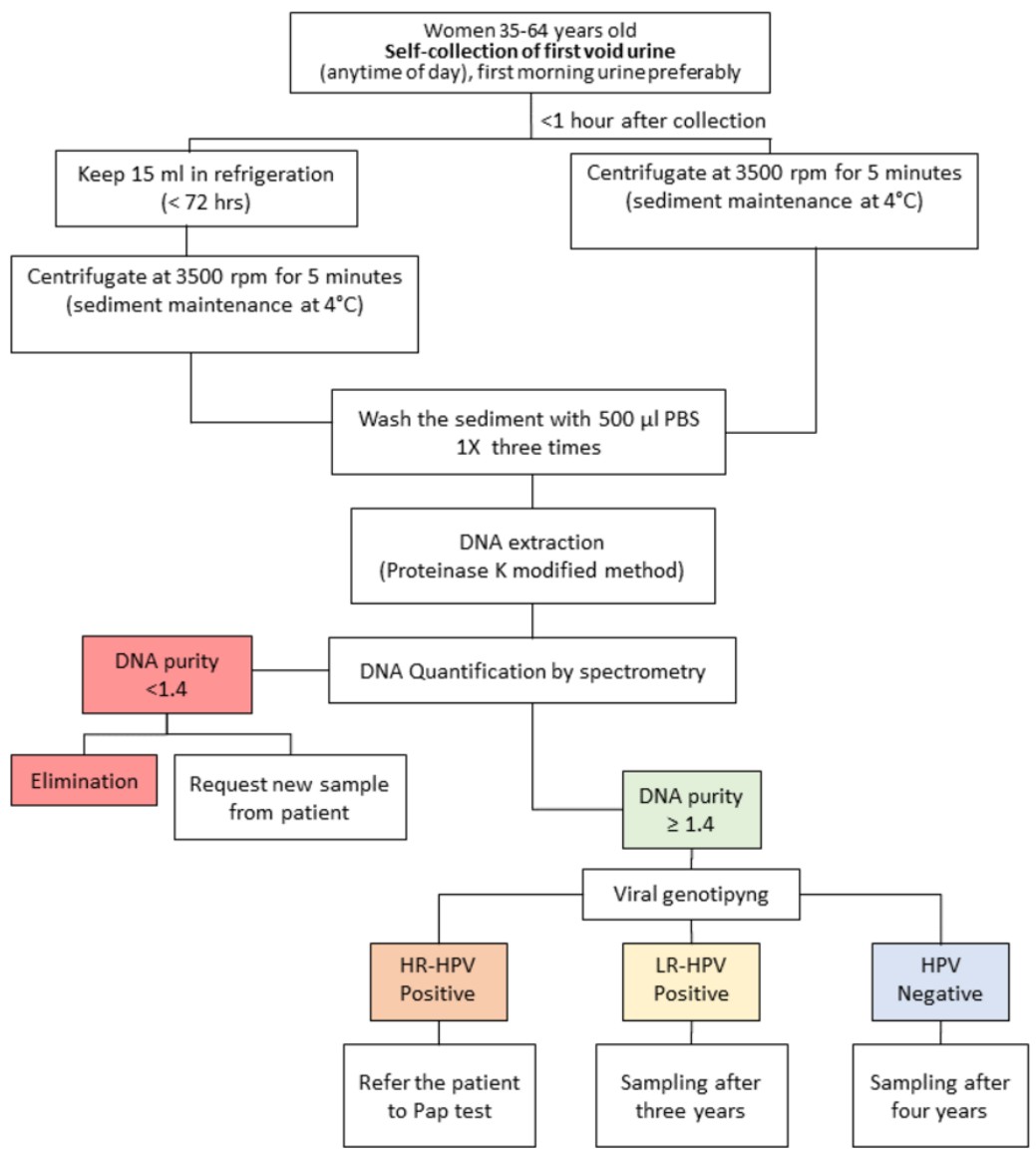

**Figure 1** **Workflow proposed for human papillomavirus (HPV) infection screening in indigenous women.** Red: The sample is not suitable for analysis; it must be discarded, and a new sample requested from the patient. Green: The sample is suitable for analysis. Blue: HPV negative, no risk of squamous intraepithelial lesions (SIL). Yellow: low risk for SIL. Orange: moderate or high risk for SIL.

## CONCLUSIONS

The use of urine samples for the molecular detection of HPV is a non-invasive method that could be a good alternative for the screening of women from indigenous populations who, for sociocultural reasons, initially refuse gynecological inspection. Only women who test positive for HR-HPV from a urine sample would be referred to gynecological inspection and the Pap test, to confirm or rule out the presence of cervical lesions and HPV infection.

## ACKNOWLEDGEMENTS

We want to thank Natividad Sales Linares for her expert technical assistance.

### Funding

This research was funded by the ''Fortalecimiento de la Investigación para el Desarrollo de la Educación y la Sociedad'' (PROFIDES) program 2017 (Grant No. PROFIDES-2017). The funders had no role in study design, data collection and analysis, decision to publish, or preparation of the manuscript.

### Grant Disclosures

The following grant information was disclosed by the authors:
''Fortalecimiento de la Investigación para el Desarrollo de la Educación y la Sociedad'' (PROFIDES): PROFIDES-2017.

### Competing Interests

The authors declare there are no competing interests.

### Author Contributions

- Francisco I. Torres-Rojas conceived and designed the experiments, performed the experiments, analyzed the data, prepared figures and/or tables, authored or reviewed drafts of the paper, and approved the final draft.
- Miguel A. Mendoza-Catalán performed the experiments, analyzed the data, prepared figures and/or tables, authored or reviewed drafts of the paper, and approved the final draft.
- Luz del C. Alarcón-Romero, Jair E. Cortes-Arciniega and Carlos J. Bracamontes-Benítez performed the experiments, prepared figures and/or tables, and approved the final draft.
- Isela Parra-Rojas conceived and designed the experiments, analyzed the data, authored or reviewed drafts of the paper, and approved the final draft.
- Sergio Paredes-Solís analyzed the data, authored or reviewed drafts of the paper, and approved the final draft.
- Marco A. Leyva-Vázquez analyzed the data, prepared figures and/or tables, authored or reviewed drafts of the paper, and approved the final draft.
- Berenice Illades-Aguiar conceived and designed the experiments, authored or reviewed drafts of the paper, and approved the final draft.

### Human Ethics

The following information was supplied relating to ethical approvals (i.e., approving body and any reference numbers):

The research protocol was approved by the Ethics Committee of the Universidad Autónoma de Guerrero (03/07/2019).

## Data Availability

The raw data used for statistical analysis are available in the Supplemental File.

## Supplemental Information

Supplemental information for this article can be found online at http://dx.doi.org/10.7717/peerj.11564#supplemental-information.

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
