# Peer review of "HPV molecular detection from urine versus cervical samples: an alternative for HPV screening in indigenous populations"

_PeerJ, doi:10.7717/peerj.11564_

## Round 0.1 · original submission · Major Revisions

Please address comments by all reviewers with a focus on improving readability.

Reviewer 1 ·

Basic reporting

The study by Torres-Rojas et al. presents concordance in the HPV detection in urine and cervical samples of 108 indigenous Mexican women. In addition, authors proposed HPV detection in urine as an alternative solution for women not attending cervical cancer. Although, present manuscript has some major shortcomings, it might be very interesting for readers and could influence policy makers. It is necessary to develop and implement alternative methods for women who are not attending cervical cancer screening as they remain at a high risk of developing cervical neoplasia. In low resource settings of indigenous populations urine or vaginal self-sampling for cervical cancer screening could be an interesting solution.
First, manuscript requires editing by professional/native English medical writer. Apart from grammatical errors throughout the manuscript it is sometimes challenging to understand long sentences, especially in the discussion section. Besides, I have some concerns regarding methodological approach and interpretation of the results/conclusions.

Could authors provide more methodological details regarding sampling and statistical analysis.
How cervical samples were taken? Which device used? How cervical samples were treated? Stored? Conventional cytology or Liquid -based? How urine samples were collected? Was a special collector used? Please provide more details. Were all samples transferred to one central laboratory for the extraction/analysis/storage etc?
How sensitivity and specificity were calculated? I assume HPV detection using INNO-LiPA was used as a reference method. Authors reported PPV, NPV and concordance but have not mentioned this in methods. I am assuming LSIL was used as a disease outcome to calculate PPV and NPV, since no biopsy was done and no histological diagnosis according to cervical intraepithelial neoplasia (CIN) stage was confirmed. As a pap smear result is not confirmed histological diagnosis it is not recommended to use it as an outcome for such a statistical measure. Moreover, LSIL is not considered as a disease and majority of LSIL cases regress to normal after 12-36 months (Moscicki, Shiboski et al. The Lancet 2004). Therefore, I strongly recommend ether to remove PPV and NPV or to address this major limitation in the discussion and provide more details in methods.
In addition, according to your results HPV detection in urine is more sensitive compared to cervical samples. I am surprised to see HPV positivity 69% and 82% in cervical and urine samples, respectively. Interestingly, the detection rate of multiple infections was two times higher in urine than in cervical samples. One would expect HPV detection in cervical samples to be more sensitive. According to supplementary table 1, HPV16 and HPV39 show highest type specific discrepancy. Could authors explain such discrepant results in the discussion?



Minor changes:
I would recommend describing in a few sentences cervical cancer screening program in Mexico.
I would suggest changing “cervical scraping” to “pap test” as sometimes cervical scaping is referred as a removal of abnormal cervical cells. Wording might be confusing to the reader.
Lines 82-87, very long sentence, please reformulate.
Ln 88 “self-take” do authors mean self-sampling, please change?
What was the proportion recruited by home visits and by calling with a loudspeaker? What is the participation rate by home visits?
Ln 135, why 1.4? Could authors please provide the reference.
Ln 153-158, most of the information could be part of table 1.
Ln 156, it is not shown % of women who do not consume alcohol.
Ln 159 – 161. Please provide exact numbers in methods.
Ln 163-166 please be consistent with decimal points.
Ln 176 please clarify in methods groupings for hr and lr HPV.
Ln 178 “as well as other HPV genotypes not identified”, do authors mean genotypes not covered by the assay? If yes, I do not think it is appropriate to mention this in results.
Ln 185 What does “MI” refer to?
Ln 186-188. The sentence is part of the discussion rather than results.
Ln 190. “Physicochemical characteristics of urine were analyzed”, How this analysis was performed? Please provide more details in methods.
Ln 203-214. I suggest moving the paragraph to the discussion.
Ln 203-206. “the presence of HR-HPV in the urine samples means that the virus is colonizing the vagina and reached the sample by entrainment, which increase the risk that papillomavirus colonize the cervical region at some point.”
Please provide the reference for the statement or remove it.
Presence of HPV in urine indicates that HPV has already infected cervicovaginal cavity. During the urination, first void urine is contaminated with exfoliated cells and mucus (containing HPV) from vagina and cervix. Hence, it is strongly recommended to use first void urine for HPV detection as it contains most of the cervical cells and, therefore, most of the viral DNA(Vorsters, Van Damme et al. BMJ : British Medical Journal 2014).

Ln 209-211. LSIL is not considered a diseases, please change to cervical intraepithelial neoplasia or cervical cancer.
Ln 219-225. Please reformulate, very long sentence.

Table 1, to show the difference between the groups I would suggest using row % rather than col.
As the sample size is very small, I would suggest to combine some categories; for example, number of sexual partners two and there versus one. Menarche ≥ 17 has only two participants, why to keep it separately etc… Please apply to other categories too.
I believe “Caesarea” is in Spanish, please change to C-section.
Table 2. please explain in footnotes what abbreviation mean HPV-LR,HPV-HR, MI HPV-LR/HR
MI HPV-HR , HPV-X. I am not sure I understand what HPV-X means and how it could be detected. For calculation proportions it would be better to separate multiple infections and hr/lr.
Please add total HPV positivity for both urine and cervix. I am not convinced that MI HPV-LR/HR and MI HPV-HR categorization adds value for the table as below authors show multiple infections. What was the range of the number of genotypes detected in urine and cervical samples?
It is interesting that number of multiple genotypes in urine is double compared to cervical samples. Please add statement in discussion. Could it be due to contamination? Did authors include negative and positive controls? How cervical samples were stored and treated? Could it be that DNA is more degraded in cervical samples?

Table 3,4 please add total values for urine HPV detection. Again, it is not recommended to calculate PPV on cytological but histological diagnosis. In addition, as LSIL regression rate is quite high I do not see the added value of PPV and NPV here.

Figure 1. In methods, you indicated that DNA purity in urine should be above 1.4, while figure indicates 1. Please be consistent and provide the reference or explain why 1.4 or 1.

According to European guidelines screening interval after negative cervical test should be at least five years. Therefore, one year screening interval would cause massive over screening. Moreover, women bellow 30-35 years old should not be screened with HPV DNA test. I would recommend to review the figure and adjust the screening interval and age. Finally, low risk HPV detection is not recommended for screening as it could cause over screening and more harm in terms of anxiety and psychological impact rather than benefits (von Karsa, Arbyn et al. Papillomavirus Research 2015, Maver and Poljak Clin Microbiol Infect 2020).

Experimental design

See above

Validity of the findings

See above

Reviewer 2 ·

Basic reporting

The abstract is clear and presents all the information required.The use of English can be revised to make the message clearer to the reader. It is suggested the use of shorter sentences. The number of figures and tables is appropriate. Abbreviations must be described in all tables and figure.

Experimental design

This study aimed to compare the concordance in HPV detection from urine and cervical samples from an indigenous population at southern Mexico.The contribution is original and its development may have practical applications in the future using an alternative to cervical scraping. The methodology presented is accurate to address the main goals of this work.
Overall, the results of the present study are not discussed in depth. For comparative purposes, the authors should include more information of HPV detection methodologies in urine.

Validity of the findings

The manuscript addresses a very important public problem.The use of urine samples for HPV genotyping is a non-invasive method that would be a good alternative for the screening of women from indigenous populations that refuse gynecological inspection.
Authors should discuss the costs of implementing urine collection and HPV genotyping using the methodology described.

Additional comments

no comment

---

## Round 0.2 · accepted · Accept

The reviewers have unanimously accepted the changes.

Reviewer 1 ·

Basic reporting

Authors addressed reviewers concerns and the manuscript has significantly improved.

Experimental design

See above

Validity of the findings

see above

Reviewer 2 ·

Basic reporting

The authors have done a god job in addressing all the comments.

Experimental design

The authors have done a god job in addressing all the comments.

Validity of the findings

The authors have done a god job in addressing all the comments.